# Determining the metabolic effects of dietary fat, sugars and fat-sugar interaction using nutritional geometry in a dietary challenge study with male mice

Jibran A. Wali [1,2] ✉, Duan Ni[1,3], Harrison J. W. Facey[1], Tim Dodgson[1,2], Tamara J. Pulpitel[1,2], Alistair M. Senior [1,2,4], David Raubenheimer[1,2], Laurence Macia [1,3,5] & Stephen J. Simpson [1,2] ✉

The metabolic effects of sugars and fat lie at the heart of the "carbohydrate vs fat" debate on the global obesity epidemic. Here, we use nutritional geometry to systematically investigate the interaction between dietary fat and the major monosaccharides, fructose and glucose, and their impact on body composition and metabolic health. Male mice (n = 245) are maintained on one of 18 isocaloric diets for 18–19 weeks and their metabolic status is assessed through in vivo procedures and by in vitro assays involving harvested tissue samples. We find that in the setting of low and medium dietary fat content, a 50:50 mixture of fructose and glucose (similar to high-fructose corn syrup) is more obesogenic and metabolically adverse than when either monosaccharide is consumed alone. With increasing dietary fat content, the effects of dietary sugar composition on metabolic status become less pronounced. Moreover, higher fat intake is more harmful for glucose tolerance and insulin sensitivity irrespective of the sugar mix consumed. The type of fat consumed (soy oil vs lard) does not modify these outcomes. Our work shows that both dietary fat and sugars can lead to adverse metabolic outcomes, depending on the dietary context. This study shows how the principles of the two seemingly conflicting models of obesity (the "energy balance model" and the "carbohydrate insulin model") can be valid, and it will help in progressing towards a unified model of obesity. The main limitations of this study include the use of male mice of a single strain, and not testing the metabolic effects of fructose intake via sugary drinks, which are strongly linked to human obesity.

The metabolic effects of dietary fats and sugars have been an area of great interest in obesity research[1–4]. Carbohydrates are the most abundant macronutrient typically accounting for 45–70% daily energy in human diets[1–3]. Fats are the macronutrient with the highest energy density, being ~2 fold more energy dense than protein and carbohydrates[2–6]. The traditional 'single nutrient approach' to nutrition science has led to controversies around the metabolic consequences of consuming fats and sugars, leading to strong debate over the roles of fats and carbohydrates in causing obesity[2,3,7]. Two competing models have been proposed to explain the causes of the global

---

obesity epidemic[8,9]. According to the conventional 'energy balance model' (EBM), obesity arises from an imbalance between calorie intake and energy expenditure, regardless of the macronutrient source of ingested energy[9]. Modern industrialised food systems are replete with highly palatable, energy-dense, ultra-processed foods, which are low in protein and fibre, facilitating consumption of excess calories and their subsequent storage as body fat[3,9–12]. In contrast, the carbohydrate insulin model (CIM) suggests that obesity is specifically exacerbated by excessive consumption of refined carbohydrates, especially caloric sugars[8,13]. These carbohydrates have a high glycaemic index, which leads to a rapid rise in postprandial blood glucose levels that strongly stimulates insulin secretion from the pancreas. Insulin directs the partitioning of the nutrients absorbed after digestion of food towards storage in adipose tissue. This hyperinsulinaemic response leads to a rapid decline in circulating glucose levels that starves the metabolically active peripheral tissues of this vital source of cellular energy[8]. As a result of this internal starvation effect, appetite signals are activated that further promote food intake and increase the availability of energy for storage in adipose tissue[8,13].

Both the EBM and the CIM thus consider caloric sugars to be important drivers of obesity[8,9]. However, these two models disagree on the role of dietary fat. The EBM suggests that because of their high energy density, hedonic properties and effects on appetite signalling, dietary fats promote obesity[9,14]. In contrast, the CIM proposes that replacing dietary carbohydrates with fats reduces postprandial glycaemic and insulinemic responses, which diminishes energy storage in adipocytes[2,8,14]. However, ultra-processed foods are often rich in both sugars and fats[15], and the presence of fat increases the energy density of foods[6], making it difficult to disentangle the metabolic effects of fat per se from the effects of excess calorie and carbohydrate intake. Therefore, a multi-nutrient approach is needed to investigate the metabolic effects of the fat-sugar interaction and to distinguish the impact of fat as a nutrient from its caloric density.

In our recent work on mice, we used nutritional geometry (NG) to investigate the apparent contradictions around different types of carbohydrates and protein-carbohydrate interaction on metabolic health[16]. We showed that the health effects of a low protein-high carbohydrate diet are dependent on the *type* of carbohydrate consumed[16]. Starch (a polymer of glucose), sucrose (a disaccharide of glucose and fructose), and high-fructose corn syrup (HFCS; most commonly a ~1:1 mixture of monosaccharide glucose and fructose) are the key sources of carbohydrate energy in modern food systems[1,17]. We found that a low protein-high carbohydrate diet can be metabolically very harmful if HFCS is the main source of carbohydrates, but low protein-high carbohydrate diets are metabolically the best if significant proportions of carbohydrate energy is provided in the form of 'resistant starch' (a type of fibre)[16].

In the present study, we aimed to use NG to investigate the metabolic consequences of fat-sugar interaction and evaluate if the metabolic outcomes align with the tenets of the CIM or the EBM model. Contrary to our previous work, where fat was fixed at 20% of total energy[16], we fixed protein at 20% in the current study (standard for mouse diets[18]) and used NG to investigate the impact of variations in dietary fat content and its interactions with caloric sugars (glucose, fructose, and their mixtures) on metabolic outcomes. A total of 245 mice were maintained on one of 18 isocaloric diets containing either lower, medium or higher fat content. Carbohydrates comprised glucose or fructose or their combinations and a fixed level of starch. We also investigated the metabolic consequences of the source of dietary fat (plant vs animal). Our results demonstrate that considering the complete dietary context can clarify some of the apparent contradictions in the EBM and CIM models that are fuelling the 'fat vs carbohydrate' debate over the rising prevalence of global obesity.

## Results

### Study design

Mice were fed *ad libitum* on one of 18 isocaloric (~14.3 kJ/g) diets with fixed protein content (20% energy) but either lower (10% fat: 70% carbohydrate), medium (20:60) or higher (30:50) fat to carbohydrate ratio. Diets were maintained isocaloric by adjusting the content of cellulose which is a commonly used strategy in rodent studies[16,19,20]. In all diets, native wheat starch constituted 30% of the carbohydrate energy, and the remaining 70% came from fructose, glucose or their combinations (fructose:glucose, 100:0, 75:25, 50:50, 25:75, 0:100). Thus, dietary fructose levels increased with decreasing glucose level, and vice versa. Overall, total fat content increased with decreasing carbohydrate content, and the presence of starch prevented fructose malabsorption, as we reported previously[16]. This experimental design facilitated the investigation of the effect of fructose, glucose and fat and their interactions using the NG platform. In 15/18 diets spanning the full range of fat:carbohydrate and fructose:glucose combinations tested, the dietary fat was sourced from soy oil. The remaining 3/18 diets varied in fructose:glucose, but were fixed at 20:60 fat:carbohydrate with lard as the fat source (Supplementary Data 1). We note that the energy-dense 'high fat diets' commonly used for inducing obesity in rodents contain 45–60% fat[5]. In contrast, the maximum fat content in our isocaloric diets was kept at 30% (noting that the standard AIN93G rodent diet contains ~20% fat energy[18]). This allowed enough scope in the dietary carbohydrate compartment to use amounts of fructose and glucose that could induce detectable phenotypic changes[16]. Further, the range of 10% to 30% fat proved sufficient to induce metabolic impacts of increasing dietary fat content. Animals were maintained on experimental diets for 18–19 weeks, and in vivo metabolic parameters were analysed after 5–6 and 12–14 weeks of dietary intervention (or as specified) (Fig. 1a). The interpretation of NG response surfaces is described in detail in the supplementary information, and elsewhere[16,21].

### The highest energy intake occurred on diets with 50:50 fructose:glucose

Average daily energy intakes were analysed at 5–6 weeks and 12–14 weeks. The major driver of food and energy intake was the ratio of fructose to glucose in the diet, with maximum intake observed on diets containing them in equal parts (50:50). Intake was lower on diets containing glucose or fructose in isolation. Reducing dietary fat content further increased total energy intake in the first few weeks of the dietary intervention, but the effect of fat was not statistically significant in the longer-term (Fig. 1b and Fig. S1a, Supplementary Data 2, 5). Note that 50:50 fructose:glucose translated into 3.5, 3.0 and 2.5 kJ/g of energy from each monosaccharide for 10%, 20% and 30% fat diets (Fig. 1b).

### Co-ingestion of fructose and glucose led to maximal body weight and adiposity

Commensurate with the intake data, mice consuming 50:50 fructose:glucose had the highest body weights, mainly due to greater fat mass (Fig. 1c, d and Fig. S1b–h, Supplementary Data 2 and 5). Gonadal and inguinal white fat pad weights were highest in mice ingesting equal amounts of fructose and glucose (Fig. 1e and Fig. S2a, Supplementary Data 2 and 6). Increasing fat intake blunted the impact of a 50:50 fructose-glucose ratio, and the body weights and adiposity of mice with the highest fat intake were minimally influenced by the ratio of fructose and glucose eaten. The body weight and fat mass of the mice with the highest fat intakes were similar to those ingesting lower quantities of fat and 50:50 fructose:glucose, and higher than those consuming lower amounts of fat and only glucose or only fructose. This was accompanied by higher gonadal and inguinal fat pad weights

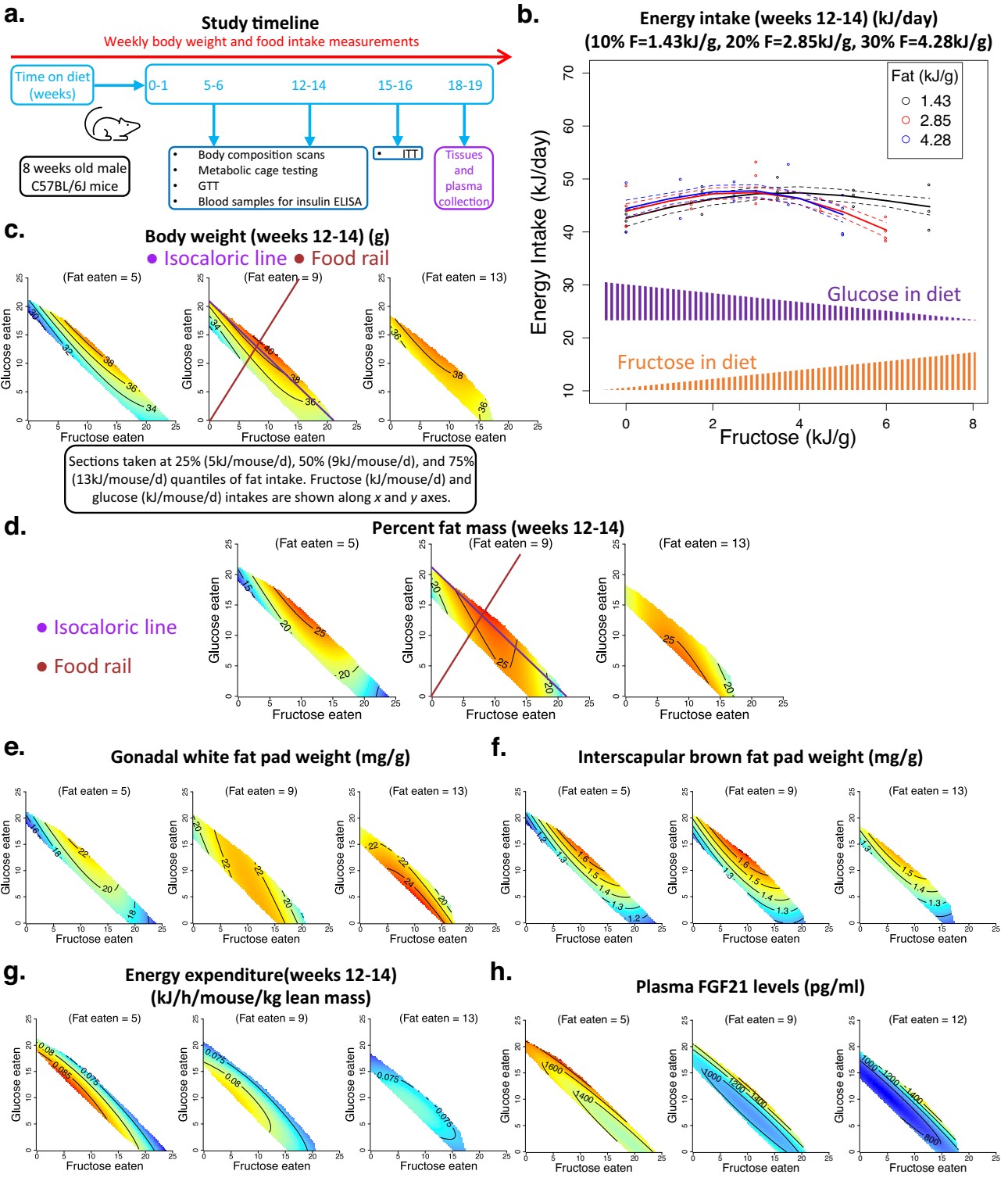

with increasing fat intake (Fig. 1c–e, Fig. S1b–h and Fig. S2a, Supplementary Data 2, 5 and 6).

The purple line in Fig. 1d intersects the highest per cent fat mass and represents isocaloric energy intake but with changing fructose:glucose ratio. On the other hand, the brown line also intersects the peak fat mass but shows the impact of increasing total energy intake across diets at a fixed fructose:glucose ratio. Visual inspection of these purple and brown lines shows that in this study, even at a fixed calorie intake, the ratio of fructose and glucose eaten strongly influenced adiposity.

**Energy expenditure and circulating FGF21 concentrations decreased with increasing fat intake**

Concurrent with differences in total fat mass, interscapular brown was highest in mice eating 50:50 fructose:glucose, and energy expenditure (measured by indirect calorimetry and normalised to lean mass) declined with increasing fat intake (Fig. 1f, g and Fig. S2c, Supplementary Data 2 and 6). This decrease in energy expenditure could be partly responsible for a generalised increase in body weight and adiposity with increasing fat consumption (Fig. 1c–e, Supplementary Data 2). Of note, absolute energy

**Fig. 1 | The effects of fat, fructose and glucose intake on body composition of mice.** (See Supplementary Data 2 for statistics). Source data are provided as a Source Data file. **a** Overall study timeline. Mice were fed on experimental diets for 18–19 weeks. After 5–6 and 12–14 weeks, metabolic parameters were measured. GTT glucose tolerance test, ITT insulin tolerance test. **b** Plot showing the effect of dietary fructose (kJ per g food) on energy intake (kJ per mouse per day) at lower (10% energy), medium (20%) and higher (30%) fat (F) content at 12–14 weeks. Along the x-axis, as fructose levels increase, glucose content in the diet decreases. For diets with a 50:50 fructose:glucose ratio, each monosaccharide was supplied at 3.5, 3.0 and 2.5 kJ per g of diet for the 10%, 20% and 30% fat diets, respectively. Each symbol (o) represents the average energy intake per mouse per cage (*n* = 4 mice per cage). The fitted lines were derived from generalised additive modelling

(GAM), fitting an interaction between a smooth term for fructose content (in one dimension) and fat content as a three-level categorical factor, and the dotted lines represent s.e.m. for fitted values. **c**–**h** Response surfaces showing the relationship between the intake of fructose-, glucose- and fat-derived energy (kJ per mouse per day) and body weight (g) (**c**), and per cent fat mass of mice at weeks 12–14 (**d**), gonadal fat pad weight (mg/g body weight) (**e**), interscapular brown fat pad weight (mg/g of body weight) (**f**), average energy expenditure over 24 h at weeks 12–14 (kJ per hour per mouse per kg lean mass) (**g**) and plasma FGF21 (pg/ml) (**h**) concentration at weeks 18–19 (**h**). Three two-dimensional (2D) slices were taken at 25%, 50% (median) and 75% quantiles of fat intake and show all three nutrient dimensions (fructose, glucose, and fat). See Supplementary Information for the details of how to interpret these NG surfaces.

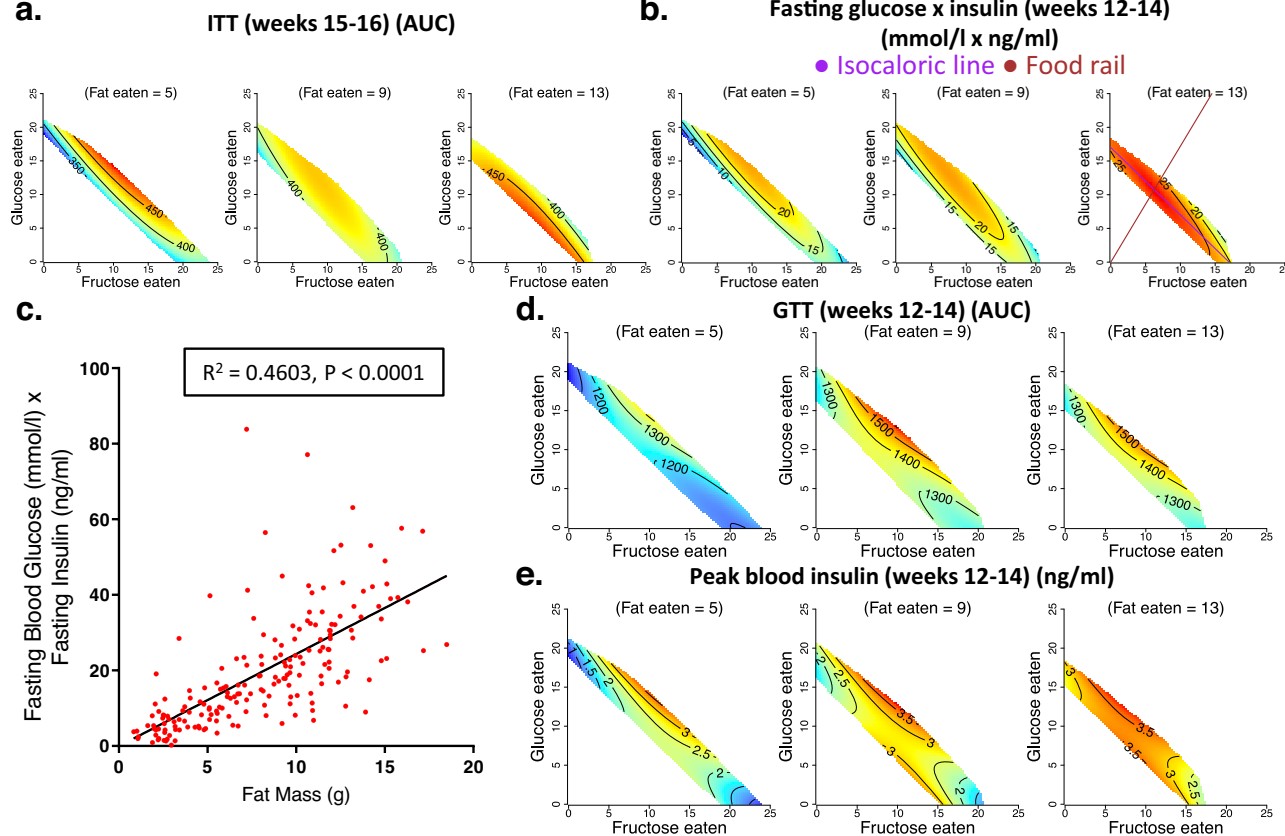

**Fig. 2 | The effects of fat, fructose and glucose intake on the glycaemic status of mice.** (See Supplementary Data 3 for statistics). Source data are provided as a Source Data file. **a**, **b**. Response surfaces showing the relationship between the intake of fructose-, glucose- and fat-derived energy (kJ per mouse per day) and insulin tolerance (AUC) at weeks 15–16 (**a**) and the product of fasting blood glucose and fasting blood insulin concentrations (mmol/l × ng/ml) at weeks 12–14 (**b**). Increasing values of the AUC for the insulin tolerance test (ITT) indicate decreasing

insulin sensitivity. **c** Relationship between fat mass (g) and the product of fasting blood glucose and fasting blood insulin concentrations (mmol/l × ng/ml) at weeks 12–14 (*n* = 193 mice). $R^2$ and *p*-value (for the regression slope, $P = 2.229 \times 10^{-27}$) for linear regression of data are shown. **d**, **e** Response surfaces showing the relationship between the intake of fructose-, glucose- and fat-derived energy (kJ per mouse per day) and the total AUC of the glucose tolerance test (GTT) at weeks 12–14 (**d**) and the peak blood insulin (ng/ml) at weeks 12–14 (**e**).

expenditure and physical activity were not significantly affected by differences in fat, fructose and glucose intake (Fig. S2b, d, Supplementary Data 6). Respiratory quotient showed an expected increase with increasing glucose and decreasing fat intake (Fig. S2e, Supplementary Data 6). The hepatokine, fibroblast growth factor (FGF21), is known to increase energy expenditure[22], and its circulating concentrations increased with increasing total carbohydrate and decreasing fat intake (Fig. 1h, Supplementary Data 2). Glucose intake was relatively more potent in increasing FGF21 levels than fructose (Fig. 1h, Supplementary Data 2).

### Co-ingestion of fructose and glucose and high fat intake impaired glucose homoeostasis

Insulin sensitivity was assessed by insulin tolerance tests and by measuring the product of fasting blood glucose and fasting blood insulin concentrations (similar to HOMA-IR in humans). Ingestion of a 50:50 fructose:glucose combination reduced insulin sensitivity (Fig. 2a, b and Fig. S3a–e, Supplementary Data 3 and 7), and fat mass significantly correlated with reduced insulin sensitivity (Fig. 2c and Fig. S3f). Whereas on low and medium fat intake, insulin sensitivity was relatively worse in mice consuming a 50:50 mixture of fructose and glucose, high fat intake produced a generalised impairment of insulin sensitivity

irrespective of the ratio of fructose to glucose consumed (Fig. 2a, b and Fig. S3a–e, Supplementary Data 3 and 7). At weeks 5–6, high carbohydrate intake more strongly increased the product of fasting glucose and fasting insulin than did high fat intake. However, at weeks 12–14, this pattern was reversed, suggesting that high fat intake is more harmful to fasting glycaemia and insulinaemia in the longer term (Fig. 2b and Fig. S3d, e, Supplementary Data 3 and 7). Moreover, animals consuming a 50:50 mixture of fructose and glucose in combination with a high fat intake showed the worst glucose tolerance and peak blood insulin concentrations in response to oral administration of a glucose bolus (Fig. 2d, e, and Fig. S3g, h, Supplementary Data 3, 7). It is unlikely that in mice with higher fat intake, the greater GTT area under the curve (AUC) and peak blood insulin concentrations after administration of a glucose bolus merely reflects their chronic exposure to relatively lower carbohydrate diets. This is because even the fasting blood insulin levels were higher in these mice consuming diets with higher fat-lower carbohydrate content (Fig. S3c, Supplementary Data 7).

### Co-ingestion of fructose and glucose increased hepatic fat content; high fat intake reduced the expression of de novo lipogenic genes

Mice co-consuming glucose and fructose, particularly at a ratio of 50:50, showed increased hepatic fat content. This was evident both when hepatic fat content was measured by a biochemical assay and on histological assessment of liver tissue (Fig. 3a–c, Supplementary Data 4). Interestingly, increasing fat intake had a minimal effect on hepatic fat deposition (Fig. 3a–c, Supplementary Data 4).

Fructose metabolism in the liver by the enzyme ketohexokinase (KHK) induces the expression of de novo lipogenic (DNL) genes[4,23]. We examined the expression of KHK (the major isoform 'C' that has higher fructose affinity[23]) and downstream DNL genes in liver tissue. High fructose intake produced an expected increase in hepatic *Khk* expression, which was most pronounced in mice with the lowest fat and the highest total carbohydrate intake (Fig. 3d, Supplementary Data 4). Similarly, genes associated with fatty acid synthesis (*Acly* and *Fasn*) were expressed at relatively higher levels in mice with higher fructose intakes (Fig. 3e, f, Supplementary Data 4). Interestingly, the expression of DNL gene *Scd1* and glycerol synthesis pathway gene *Gpat3* were highest in mice on 50:50 fructose:glucose (Fig. S4a, b, Supplementary Data 8). This suggests that possibly due to the increase in *Gpat3* and *Scd1* expression, liver triglyceride content peaked in mice ingesting 50:50 fructose:glucose and not in mice with the highest fructose intakes. We observed a decrease in the expression of lipogenic genes with increasing dietary fat intake (Fig. 3d–f, Supplementary Data 4). This indicates that de novo fat synthesis in the liver decreases as dietary fat intake increases and is consistent with previous findings[24]. Moreover, the gene expression of Apolipoprotein-B (*Apob*), which is involved in exporting lipids out of the liver into circulation[25], also decreased with high fat intake (Fig. S4c, Supplementary Data 8). In contrast, the hepatic expression of the pro-inflammatory gene *Mcp1* was highest in mice with the highest fat intake (Fig. 3g, Supplementary Data 4). The patterns of lipogenic gene expression in the liver did not translate into significant changes in circulating triglyceride concentrations (Fig. S4d, Supplementary Data 8). This is consistent with our previous observations that protein intake is the key determinant of triglyceridaemia[16,19].

### Replacing soy oil with lard fat did not alter metabolic outcomes

In the experiments described above, dietary fat was sourced from plant-based soy oil. Plant oils have a relatively lower saturated fat content (especially palmitic acid) than animal fats[26]. Therefore, we investigated if replacing soy oil with lard altered the nature of fat-fructose-glucose interaction and the associated metabolic outcomes. Mice were fed diets containing 20% protein, 20% fat and 60%

carbohydrates, with their fat sourced from either soy oil or lard. Dietary carbohydrates comprised 30% native wheat starch, with the remaining 70% carbohydrate energy sourced from either glucose or fructose or their 50:50 mixture (Supplementary Data 1).

For both soy-based and lard-based diets, diets containing a 50:50 ratio of fructose and glucose led to the highest body weights, absolute and per cent fat mass, and interscapular brown fat pad weight (Fig. 4a, b, Fig. S5a, b), while absolute lean mass was similar across the diets (Fig. 4c). However, the source of fat (soy oil vs lard) did not make any statistically significant difference to body weight, body composition and energy intake (Fig. 4a–d). Similarly, data for insulin sensitivity, glucose tolerance, peak blood insulin and liver triglyceride content showed that mice fed diets containing 50:50 fructose and glucose were metabolically the worst but replacing soy oil with lard had no effect (Fig. 5a–d). Plasma triglyceride concentrations were also similar in mice-fed diets containing either soy oil or lard (Fig. S5c).

## Discussion

In this study, we used nutritional geometry to investigate how the dietary fat-sugar interaction influences metabolic status and if the consequences of this interaction are dependent on the type of sugar (fructose vs glucose vs their mixtures) and fat (soy oil vs lard) consumed. Consistent with our previous work[16], we found that a 50:50 mixture of fructose and glucose was more obesogenic than the consumption of fructose or glucose alone. This 50:50 ratio of fructose and glucose is the same as the ratio in the disaccharide sucrose, and it is similar to the ratio found in most commonly consumed varieties of high-fructose corn syrup, HFCS-42 and HFCS-55, which contain 42% and 55% fructose, respectively[27,28]. The 50:50 fructose-glucose ratio increased body weight and adiposity by promoting greater calorie intake as well as ratio-specific effects independent of the caloric value. Similar to these results, experiments in humans showed that the decrease in appetite scores after consuming a 50:50 ratio of fructose and glucose was the lowest when compared with various other ratios of fructose and glucose[29]. Moreover, our finding that co-ingestion of fructose and glucose led to maximum hepatic fat content is supported by observations in humans and mice where co-ingestion of fructose and glucose was shown to strongly induce de novo lipogenesis in the liver compared with consumption of individual monosaccharides[16,30,31].

The level of fat intake influenced the metabolic effects of consuming a 50:50 mixture of fructose and glucose. At low-to-medium fat intakes, body weights and adiposity were highest in mice consuming the 50:50 fructose-glucose mixture, but at higher fat intakes, the body weights of animals consuming 50:50 fructose-glucose mixture became very similar to those consuming only fructose or only glucose. Thus, a high fat intake caused a more generalised increase in adiposity and body weight that was largely independent of the type of sugar in the diet. In addition, compared with the consumption of a 50:50 mixture of fructose and glucose, a higher fat intake more adversely affected fasting insulin levels, insulin sensitivity and glucose tolerance.

Our results show how certain aspects of both the EBM and the CIM models of obesity could be valid depending on the dietary context. For example, supporting the EBM model, mice with the highest energy intakes, which were achieved on diets containing 50:50 fructose and glucose, had the highest body weights and adiposity. Moreover, a higher fat intake caused a greater increase in fasting insulinemia and more adversely affected glucose tolerance than a higher carbohydrate intake. These results are contrary to the outcomes that would have been expected from the CIM model. For example, the glycaemic index of glucose, fructose and HFCS-55 is 100, 19 and 58, respectively[32]. Therefore, according to CIM, diets containing 100% glucose would have produced a higher postprandial insulinemic response and should have led to greater weight gain. But supporting the CIM, higher carbohydrate intake in the form of 50:50 mixtures of fructose and glucose led to greater liver fat content, and this was minimally affected by an

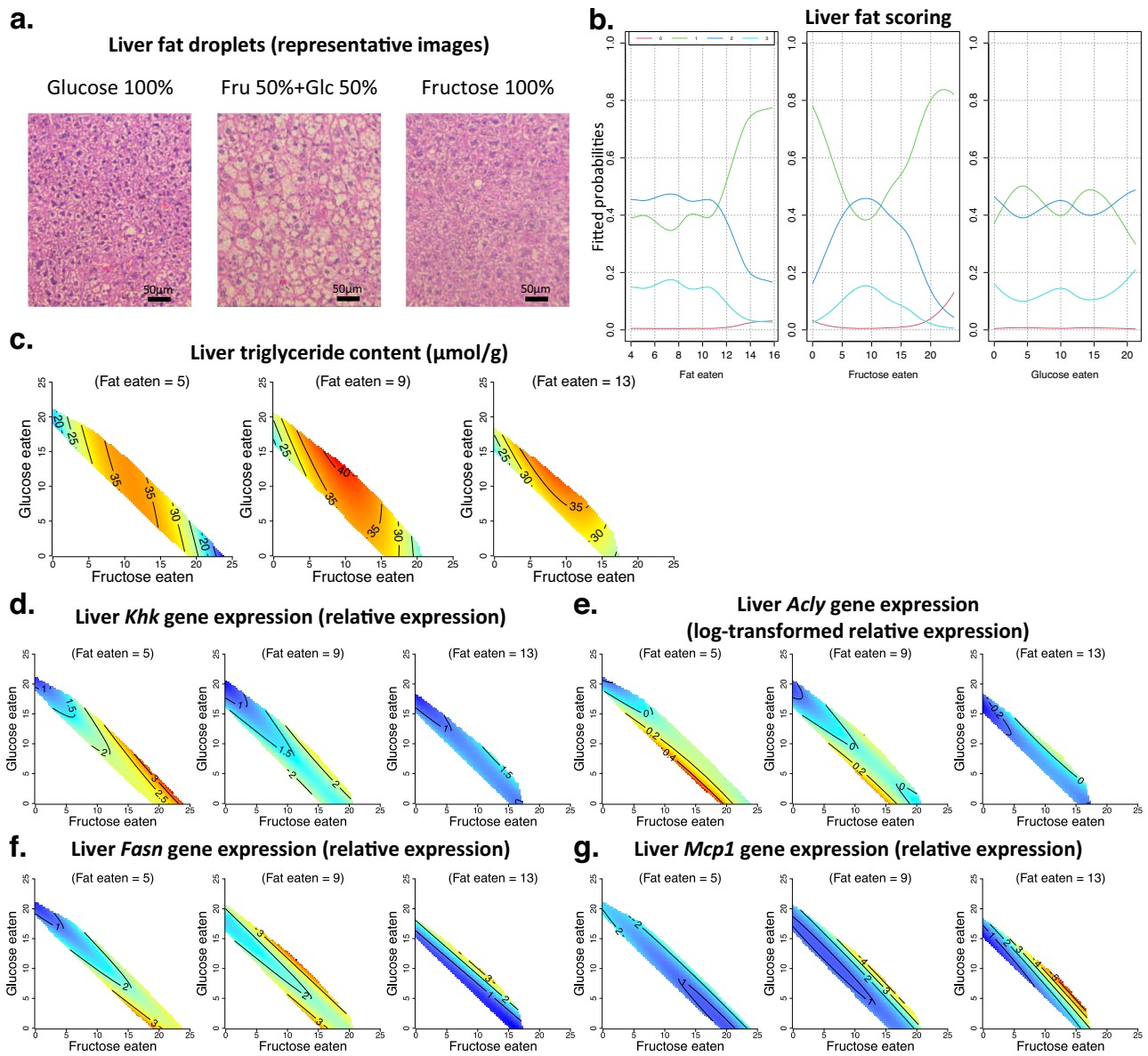

**Fig. 3 | The effects of fat, fructose and glucose intake on liver metabolism.** (See Supplementary Data 4 for statistics). Source data are provided as a Source Data file. **a**, **b**. Formalin-fixed sections of liver tissue from mice fed on experimental diets for 18–19 weeks were stained with haematoxylin and eosin (H&E) and scored for the presence and severity of liver fat. **a** Representative images of mice (n = 6 mice per diet) fed on diets with 20% fat content and monosaccharides fructose (100% fructose (Fru)) or glucose (100% glucose (Glc)) or a 1:1 combination (50% fructose and 50% glucose) are shown. **b** Relationship between the intake of fructose-, glucose- and fat-derived energy (kJ per mouse per day) and the risk of increased hepatic fat deposition (fitted probabilities) is shown. Slides were scored for liver fat (0 (red line), 1 (green line), 2 (blue line) and 3 (light blue line)), with increasing scores reflecting greater liver fat deposition. **c** Response surfaces showing the relationship between the intake of fructose-, glucose- and fat-derived energy (kJ per mouse per day) and liver triglyceride content (μmol triglyceride per g tissue) in mice fed on experimental diets for 18–19 weeks. **d**–**g**. Response surfaces showing the relationship between the intake of fructose-, glucose- and fat-derived energy (kJ per mouse per day) and the relative expression of fructose metabolism gene *Khk* (**d**), de novo lipogenesis pathway genes *Acly* (**e**) and *Fasn* (**f**) and inflammation pathway gene *Mcp1* (**g**), in liver tissue harvested at weeks 18–19.

increase in fat intake. In addition, partly consistent with CIM, energy intake itself was increased primarily by diets with 50:50 fructose to glucose and was not significantly affected by dietary fat content.

Our work sheds light on how both low fat-high carbohydrate and high fat-low carbohydrate diets could reduce obesity[33–35]. Low fat-high carbohydrate diets containing HFCS as the major carbohydrate would be predicted to cause obesity by facilitating greater calorie intake. However, low fat-high carbohydrate diets could be more effective than low carbohydrate-high fat diets in reducing ad libitum energy intake and inducing loss of fat mass if carbohydrate is not consumed in the form of fructose-glucose mixtures. This has been confirmed in a recent

human study where a low-fat diet led to lower calorie intake and a greater decrease in adiposity than a ketogenic diet[36]. In contrast, the results of the present study suggest that minimal improvement in metabolic status is to be expected if dietary fat is replaced with HFCS or sucrose. Therefore, unsurprisingly, the prevalence of obesity continued to increase as dietary fats were replaced by processed caloric sugars over the last few decades[3,37].

Two aspects of our data on dietary fat and its metabolic effects warrant further comment. First, similar to other reports where the dietary fat-carbohydrate ratio was altered in isocaloric settings[19,20], increasing dietary fat content and fat-to-carbohydrate ratio did not alter

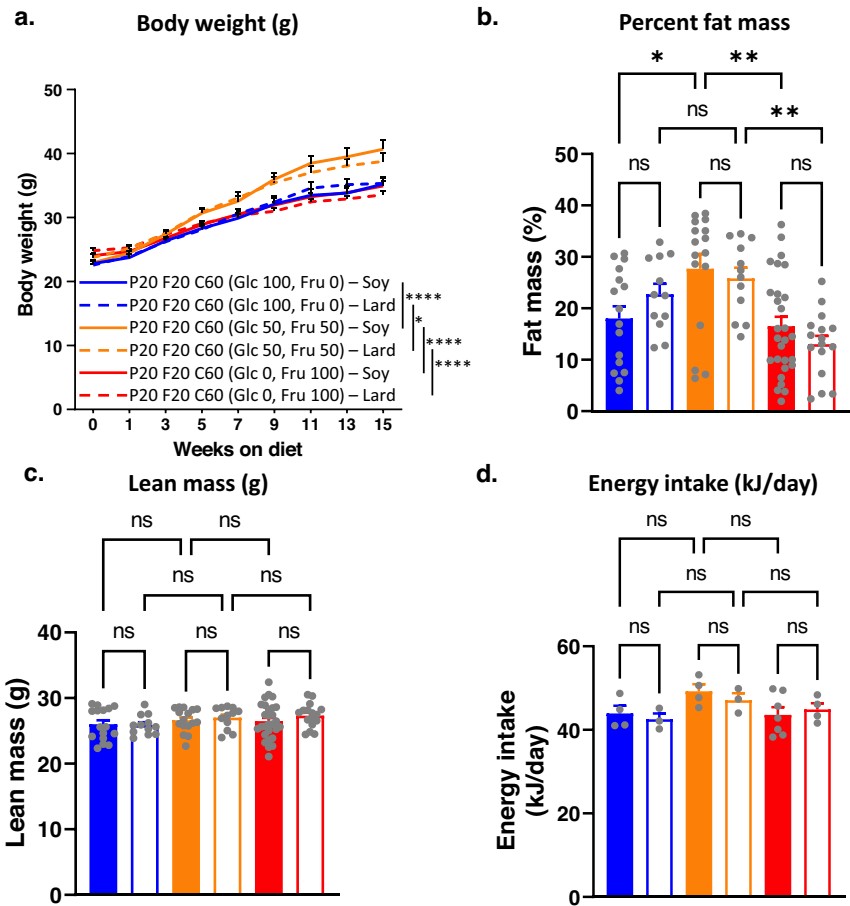

**Fig. 4 | The effects of substituting soy oil with lard as the dietary fat source on metabolic phenotype.** Source data are provided as a Source Data file.
**a** Bodyweight (g) of mice fed on diets with fat sourced from either soy oil (S) or lard (L) and carbohydrate sourced from monosaccharide fructose (100% fructose (Fru)) (F100S (n = 28) or F100L (n = 16)) or glucose (100% glucose (Glc)) (G100S (n = 16) or G100L (n = 12)) or a 1:1 combination (50% fructose and 50% glucose) (F50G50S (n = 16) or F50G50L (n = 12)). *P < 0.05, ****P < 0.0001 (P < 0.0001 for G100S versus G50F50S, P < 0.0001 for G50F50S versus F100S, P = 0.0484 for G100L versus G50F50L, P < 0.0001 for G50F50L versus F100L) for two-way ANOVA (Tukey–Kramer post hoc test) between indicated diets. Mean + s.e.m. **b** Per cent fat mass of mice fed on experimental diets at weeks 12–14. The numbers of animals for G100S, G100L, F50G50S, F50G50L, F100S and F100L were 16, 12, 16, 12, 28 and 16, respectively. ns = not significant, *P < 0.05, **P < 0.005 (P = 0.0356 for G100S versus

G50F50S, P = 0.0019 for G50F50S versus F100S, P = 0.0045 for G50F50L versus F100L) for two-way ANOVA (Tukey–Kramer post hoc test). Mean + s.e.m. ns = not significant. Each symbol (●) represents an individual mouse. **c** Lean mass of mice fed on experimental diets at weeks 12–14. The numbers of animals for G100S, G100L, F50G50S, F50G50L, F100S and F100L were 16, 12, 16, 12, 28 and 16, respectively. No significant difference was reported from two-way ANOVA (Tukey–Kramer post hoc test). Mean + s.e.m. Each symbol (●) represents an individual mouse. **d** Average energy intake (kJ per mouse per cage per day; n = 3–7 cages per diet) of mice fed on experimental diets at weeks 12–14. The numbers of cages for G100S, G100L, F50G50S, F50G50L, F100S and F100L were 4, 3, 4, 3, 7 and 4, respectively. No significant difference was reported from two-way ANOVA (Tukey–Kramer post hoc test). Mean + s.e.m. Each symbol (●) represents an individual cage.

---

ad libitum energy intake in our study. This may, at first, seem contrary to a recent mouse study that showed an increase in ad libitum energy intake and adiposity with increasing dietary fat content[38]. However, in contrast with the present study, the diets used were not isocaloric, making it impossible to differentiate the effects of dietary caloric density from the effects of fat per se on energy intake. It is important to make this distinction because EBM argues that excess calories from all sources (including fats and sugars) are obesogenic[9,14]. Second, our expectation was that, because of their greater saturated fat content, consuming lard-based diets would be metabolically more detrimental than soy oil-based diets, especially when coupled with HFCS. However, we found that replacing soy oil with lard did not affect the metabolic phenotype of the mice. Others have also observed in rodents that the type of fat in the diet did not alter body weight and composition[39,40]. This indicates that total fat content is more important than the proportion of saturated fat for inducing detectable metabolic changes.

The main limitations of this study include: (a) only simple sugars were used, and they were not compared with complex carbohydrates

with low glycaemic index, (b) post prandial glycaemic and insulinemic response to experimental diets and the effects of insulin on glucose clearance, fat metabolism and appetite were not studied, (c) impact of diets on hypothalamic appetite and hedonic signalling was not examined, (d) diets with very high-fat content or high energy densities were not used, (e) cellulose content was adjusted to keep the diets isocaloric, (f) fructose and glucose were given only as solid diets, and not as liquid solutions, (g) metabolic effects of fat and sugars were not tested in pair feeding experiments, (h) single strain and single-sex of the mice was used, (i) mouse experiments were not repeated in thermoneutral conditions. The aim of this study was to interrogate the consequences of fat-sugar interaction. Thus, we used the monosaccharides found in major caloric sugars (i.e., glucose and fructose). We have already compared the metabolic effects of simple sugars with glucose polymers (starch and resistant starch) in our previous work[16]. Although postprandial glucose and insulin levels were not specifically examined in response to experimental diets, data shown for glucose and insulin levels from fasting state and from oral glucose tolerance

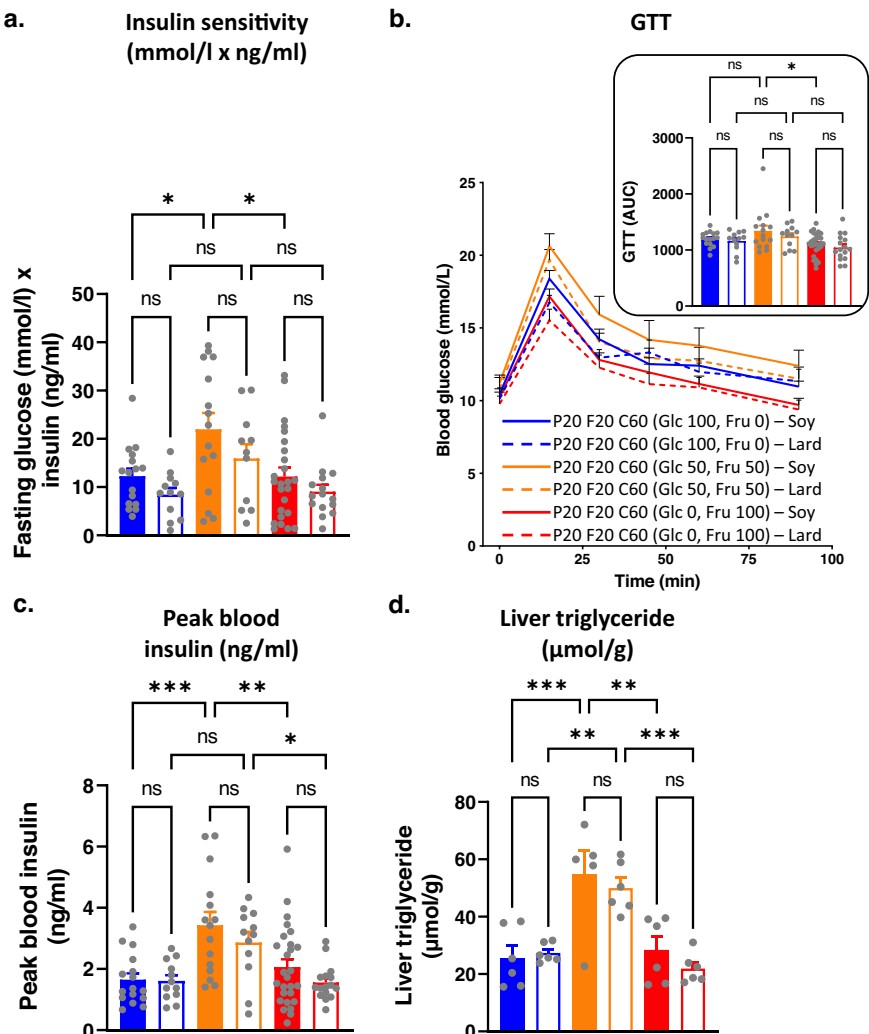

**Fig. 5 | The effects of substituting soy oil with lard as the dietary fat source on the glycaemic status of mice.** Source data are provided as a Source Data file. **a** Insulin sensitivity (fasting glucose × insulin concentration (mmol/l × ng/ml)) of mice fed on experimental diets at weeks 12–14. The numbers of animals for G100S, G100L, F50G50S, F50G50L, F100S and F100L were 16, 12, 15, 11, 25 and 15, respectively. ns = not significant, *$P < 0.05$ ($P = 0.0302$ for G100S versus G50F50S, $P = 0.0116$ for G50F50S versus F100S) for two-way ANOVA (Tukey–Kramer post hoc test). Mean + s.e.m. Each symbol (●) represents an individual mouse. **b** GTT curves and AUC of mice fed on experimental diets at weeks 12–14. The numbers of animals for G100S, G100L, F50G50S, F50G50L, F100S and F100L were 16, 12, 15, 12, 28 and 16 respectively. ns= not significant, * $P < 0.05$ ($P = 0.0182$) for two-way ANOVA (Tukey-Kramer post-hoc test). Mean + s.e.m. Each symbol (●) represents an individual mouse. **c**. Peak blood insulin of mice fed on experimental diets at weeks 12–14. The numbers of animals for G100S, G100L, F50G50S, F50G50L, F100S and F100L were 16, 12, 15, 12, 27 and 16 respectively. *$P < 0.05$, **$P < 0.005$, ***$P < 0.001$ ($P = 0.0005$ for G100S versus G50F50S, $P = 0.0043$ for G50F50S versus F100S, $P = 0.0382$ for G50F50L versus F100L) for two-way ANOVA (Tukey–Kramer post hoc test). Mean + s.e.m. Each symbol (●) represents an individual mouse. **d** Liver triglyceride content (µmol triglyceride per g tissue) in mice fed on experimental diets for 18–19 weeks. The numbers of animals for G100S, G100L, F50G50S, F50G50L, F100S and F100L were 6, 6, 5, 6, 6 and 6, respectively ns = not significant, **$P < 0.005$, ***$P < 0.001$ ($P = 0.0009$ for G100S versus G50F50S, $P = 0.0030$ for G50F50S versus F100S, $P = 0.0089$ for G100L versus G50F50L, $P = 0.0008$ for G50F50L versus F100L) for two-way ANOVA (Tukey–Kramer post hoc test). Mean + s.e.m. Each symbol (●) represents an individual mouse.

tests (Fig. 2d, e, Fig. S3b, c) is a good indicator of glucose metabolism. Further, hypothalamic appetite signalling and hedonic stimuli in response to experimental diets are most likely to reflect energy intake data (Fig. 1b). We did not use very high-fat content (>45%) in our diets for two reasons: (i) using very high levels of fat would have meant decreasing total carbohydrate content and leaving little room to study changes in carbohydrate composition, (ii) fat has higher energy density than protein and carbohydrate. Using large amounts of fat would have also meant using substantial amounts of cellulose to keep the diet isocaloric. This would have made dietary cellulose content a potentially important confounder when interpreting our data. Nonetheless, the range of fat content (10–30%) used in this study was wide enough to observe the metabolic consequences of increasing fat intake, and cellulose did not mask these outcomes. There is evidence that

fructose-containing sugars are more obesogenic when consumed via beverages[41]. Therefore, not testing metabolic outcomes when fructose is ingested in liquid form makes our findings less relevant to the role of sugary beverages in human obesity[17,42]. However, a direct experimental comparison of the metabolic effects of consuming fats and sugars in liquid form requires making mice drink large amounts of oil solutions which is challenging, and we were able to identify the ratio of fructose and glucose (50:50), which is obesogenic in the solid form. While pair feeding could provide additional information in separating the impact of nutrients from their caloric value, such studies often lead to extended periods of fasting in the pair-fed group as these mice tend to consume their food soon after it is made available[43]. Importantly, with the NG methodology, we can evaluate the impact of nutrients on metabolic phenotype at fixed levels of energy intakes (e.g., isocaloric

purple lines in Fig. 1c, d). We performed our experiments in male C57BL/6 mice, which is the most widely used animal model of diet-induced obesity. However, not repeating our experiments in females, other strains of mice and under thermoneutral conditions is an important limitation of this work. Future research should examine if the metabolic effects observed in this study are dependent on the strain and sex of the mice as well as their housing temperatures.

In conclusion, this study showed that in diets with a low-to-medium fat content, HFCS consumption led to greater food and energy intakes, body weights and adiposity when compared with consumption of glucose or fructose alone. However, with increasing fat intake, sugar-specific differences in metabolic effects became less pronounced, and there was a more generalised increase in body weight and adiposity, and greater impairment of glucose tolerance and insulin sensitivity. Using NG to disentangle the relative roles of fat and sugar intake in human data will further reconcile the differences between CIM and EBM as models of obesity.

## Methods

This study was approved by the institutional ethics committee at the University of Sydney. All animal procedures and protocols were approved by the animal ethics committee at the University of Sydney (protocol number 2018/1362).

### Interpretation of nutritional geometry surfaces
A detailed explanation of how to interpret nutritional geometry surfaces shown in the figures is available in supplementary materials.

### Animals, diets, and study design
C57BL/6J male mice (4-week-old) were purchased from the Animal Resources Centre. They were housed in 4/cage with a 12-h light/dark cycle at 24–26 °C and 44–46% humidity setting. Male mice were chosen as they are more prone to diet-induced metabolic abnormalities than females[44]. Mice were acclimatised in the animal facility for 4 weeks while being fed regular brown chow. The ad libitum dietary intervention commenced when the mice became 8 weeks old. After 18–19 weeks of dietary intervention, mice were euthanised between 1000 and 1200 h after administering an overdose of pentobarbitone (75 mg/kg body weight), and tissues and plasma were collected and biobanked. In vivo metabolic procedures, food intake, body weight measurements and animal tissue collections were performed as previously described[16]. Further details about the study design and animal numbers per diet are available in the Supplementary information.

All 18 experimental diets were isocaloric with a net metabolisable energy of ~14.3 kJ/g and were based on the AIN93G standard rodent diet (Supplementary Data 1). In terms of net metabolisable energy, the AIN93G diet contains ~19% protein, ~17% fat and ~64% carbohydrate[18]. For 15/18 experimental diets, fat was sourced from soy oil, while for the remaining 3/18 diets, lard fat was used. In all 18 diets, the protein was fixed at 20% to match the AIN93G diet. Protein was sourced from casein and carbohydrate comprised of a mixture of the sources shown in Supplementary Data 1. The diets were kept isocaloric by altering their cellulose content. All 18 diets were manufactured by Specialty Feeds™ (Glen Forrest, Western Australia) with the following catalogue numbers: SF18-090, SF18-091, SF18-092, SF18-093, SF18-094, SF18-095, SF18-096, SF18-097, SF18-098, SF18-099, SF18-111, SF18-112, SF18-113, SF18-114, SF18-115, SF18-173, SF18-174 and SF18-175.

### Body composition
MRI scanning of mice (EchoMRI™) was used to determine the body composition (fat and lean mass) of mice. Conscious mice were analysed after weeks 5–6 and 12–14 on diets.

### Metabolic cage experiments
Mice (5–7 per diet) from each diet were housed in Promethion metabolic cages (Sable Systems) for 48 h. After 24 h of acclimatisation, $O_2$ consumption, $CO_2$ production, respiratory quotient, energy expenditure, and physical activity were measured by indirect calorimetry. Physical activity was measured by cumulative y-axis beam break counts. Data were analysed by using the CalR online tool[45].

### Glucose tolerance test
Mice were orally administered with 2 g per kg lean mass of glucose after 6 h fasting. Blood was sampled from tails at baseline and 15, 30, 45, 60 and 90 min post-glucose administration to measure the blood glucose (Accu-Chek Performa, Roche). Total AUC was calculated from blood glucose readings[46]. Higher AUC indicates worse glucose tolerance. Moreover, 10 microlitres of blood were collected at baseline and at 15 min post oral glucose gavage to measure fasting and peak blood insulin concentrations.

### Insulin tolerance test
An insulin tolerance test was performed at weeks 15–16 of dietary treatment. Mice were intraperitoneally injected with 0.75 U insulin per kg lean mass (Actrapid, Novo Nordisk). Blood glucose was measured at 0, 15, 30 and 45 min post-injection. Individual AUC was calculated from blood glucose readings. Higher AUC indicates lower insulin sensitivity.

### Insulin and FGF21 ELISA
Blood samples were collected during GTTs, and insulin levels were quantified with the Ultrasensitive Insulin ELISA kit (Crystal Chem). Plasma FGF21 was quantified with the mouse FGF21 ELISA kit (Bio Vendor) for blood samples collected at the end of dietary interventions.

### Liver histology
After harvesting, livers were fixed in formalin and embedded in paraffin. 5 mm sections were stained with H&E and then scored for the presence of fat (0–3) blinded to their categories by three independent observers.

### Plasma biochemistry
Triglyceride levels in plasma samples were analysed by a clinical chemistry analyser at the Charles Perkins Centre, University of Sydney.

### Quantitative real-time PCR
30–50 mg of liver tissue was used for RNA extraction with the TRIzol method (Thermo Fisher Scientific). cDNA was then synthesised using the iScript Reverse Transcriptase enzyme and random hexamer primers (Bio-Rad). 2 ml of each RNA sample was pooled as a cDNA control sample, which was used for normalising gene expression data. cDNA was loaded in a 384-well plate format with SYBR Green (Bio-Rad) fluorescent chemistry in a total 10 microlitres reaction volume with specific forward and reverse primers. Quantitative PCR programmes were run on a Roche LightCycler (Roche) following the manufacturer's protocol. Ribosomal protein gene Rpl13a was chosen as the housekeeping gene[47] after testing a fraction of samples for gene expression of actin, Rpl13a and cyclophilin. Ct values of housekeeping and candidate genes were determined, and their expression was calculated by the DDCt method. Primer sequences for *Rpl13a*[47], *Apob*[48], *Khk* (isoform C)[49], *Gpat3*[50], *Acly*[16], *Fasn*[16], *Scd1*[16] and *Mcp1*[51] were from previous publications.

### Liver triglyceride assay
Liver triglyceride level was quantified as reported in previous studies[16,52]. In brief, 30–40 mg of liver tissue was used to extract triglyceride with a 2:1 chloroform:methanol solution. The lipid extract was dried down with nitrogen gas and resuspended in 500 ml ethanol.

A colourimetric assay was then used to quantify the triglyceride concentration with glycerol standards (Precimat glycerol, Roche) and the Triglyceride-GPO-PAP reagent (Roche).

### Statistical analysis

Details of data analysis by the NG platform and its interpretation with general additive models (GAMs) were described previously[16,19,53]. For the NG-based analysis, GAMs with thin-plate splines were used to model the responses of mice over the nutrient-intake space in R (version 4.1.1)[16,19]. Statistical outcomes retrieved from GAMs are provided in Supplementary Data. Scatter plots such as energy intake analysis were analysed by GAMs as well, fitting an interaction between a smooth term for dietary sugar content (in one carbohydrate dimension) and fat contents as three-level categorical factors and shown as scatterplots (the dotted lines on scatterplots represent the standard error for the fitted values). All GAMs underwent model validation with analysis of the residuals. Data were log-transformed if needed. For histological studies, sections were given scores ranging from 0 to 3, and the scores were modelled with an ordinal regression (proportional odds) in R. For soy oil versus lard studies, data were analysed with ANOVA in GraphPad Prism software. Data were expressed as mean ± s.e.m., and $P < 0.05$ was considered statistically significant.

### Reporting summary

Further information on research design is available in the Nature Portfolio Reporting Summary linked to this article.

## Data availability

All data supporting the findings described in this article are available in the article and in the Supplementary Information and from the corresponding authors upon reasonable request. Source data are provided in this paper.

## Code availability

Custom R scripts used for data analysis in this study were uploaded to GitHub previously and are available at: https://github.com/Nidane/Sugar-Fat-Study.

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

## Acknowledgements
J.A.W. was supported by a Peter Doherty Biomedical Research Fellowship from the National Health and Medical Research Council of Australia (GNT1125343). This work was supported by a programme grant from the National Health and Medical Research Council (GNT1149976) awarded to S.J.S., D.G.L.C. and D.R. (and their colleagues J. George, J. Gunton and H. Durrant-Whyte) and a project grant from Diabetes Australia (Y17G-WALJ) awarded to J.A.W. We thank P. Teixeira for administrative support; the Laboratory Animal Services at the University of Sydney for animal care and support; and W. Potts from the Specialty Feeds company.

## Author contributions
S.J.S., D.R. and J.A.W. conceived the study. J.A.W., D.N. and S.J.S. wrote the paper. L.M. and D.R. reviewed the paper and provided intellectual input. J.A.W., D.N., H.J.W.F., T.D. and T.J.P. conducted mouse studies. J.A.W., D.N. and A.M.S. were involved in data analysis.

## Competing interests
The authors declare no competing interests.

## Additional information

¹Charles Perkins Centre, The University of Sydney, Sydney, NSW, Australia. ²Faculty of Science, School of Life and Environmental Sciences, The University of Sydney, Sydney, NSW, Australia. ³School of Medical Sciences, Chronic Diseases Theme, Faculty of Medicine and Health, The University of Sydney, Sydney, NSW, Australia. ⁴Sydney Precision Data Science Centre, The University of Sydney, Sydney, NSW, Australia. ⁵Sydney Cytometry, The University of Sydney, Sydney, NSW, Australia. ✉e-mail: jibran.wali@sydney.edu.au; stephen.simpson@sydney.edu.au

