## [Peer Review File · Nature Communications]

Determining the metabolic effects of dietary fat, sugars and fat-sugar interaction using nutritional geometry in a dietary challenge study with male miceEditorial Note:

This manuscript has been previously reviewed at another journal that is not operating a transparent peer review scheme. This document only contains reviewer comments and rebuttal letters for versions considered at *Nature Communications*.

REVIEWERS' COMMENTS

Reviewer #1 (Remarks to the Author):

I appreciate the authors' thorough responses to my previous comments. However, it is still questionable whether their study has any meaningful implications on human diets and health consequences. The most concerning part is the fact that they used only solid fructose, which is known not as harmful as liquid fructose according to numerous human intervention studies. They raised technical issues of feeding mice with their special diet but they can at least try pair feeding or feeding restriction methods to overcome this issue. Ad libitum feeding is simple and easy for investigators but it limits the value of the study. This reviewer does not ask to repeat all their comprehensive experiments with liquid fructose, but the authors should perform a few critical experiments strategically at least to strengthen their previous conclusion or draw different conclusions. Also, I agree with the high cellulose content concern raised by another reviewer, which limits relevance to human diet. The authors mentioned they cannot measure postprandial metabolism without sacrificing mice, but measuring fed state metabolism is possible with simple blood tests at night times or early morning. Lastly, thermoneutral temperatures could affect many factors other than just brown fat effects such as eating behavior, obesogenic effect, insulin sensitivity, energy consumption, etc. How do authors know this without testing it on their dietary conditions? Overall, while this study is interesting and potentially important, this reviewer hopes to see data from experimental conditions that better mimic human diet and physiology to support the publication.

Reviewer #2 (Remarks to the Author):

This is an interesting paper that investigates the role of monosaccharides versus fat with 18 different diets that are isocaloric in calories/g of food. All of the diets varied from 50 to 70 percent carbohydrate and from 10 to 30 percent fat, with 20 percent protein in all diets. The main finding was that energy intake was greatest with a 50/50 mixture of glucose and fructose, but this was overcome to some extent by increasing the fat content. The authors argue that this supports both the CHO insulin theory and the energy balance theory.

Comments

While I agree that the data does support both models, I do think there are shortcomings. The two major ones are that the studies involve only males and one strain of mouse, and that there are very limited studies investigating mechanisms. The increased energy intake is likely from leptin resistance but could involve other mechanisms and it would be important to understand this better. A third weakness is that all of the models involved high carb diet (50-70%). There is some evidence that high CHO diets may be able to stimulate energy intake by inducing leptin resistance which is not seen with high fat diets, but in this case there were no controls with low (30% CHO) diets to really tease this out. I also agree with Reviewer 1 who argued that some postprandial studies would have been helpful, as well as my prior request to look at endogenous fructose pathways. Finally, it would have also been optimal to provide some mice with fructose/glucose in the drink, as that is much more obesogenic than putting it into the water.

Reviewer #4 (Remarks to the Author):

The editor has asked me to comment on the author's response to Reviewer #3 due to my expertise in diet design and mouse studies. I have read the paper and the reviews, and I think that this manuscript

by Simpson and colleagues is interesting, reports novel results that will be of interest to a broad audience of scientists, as well as to the public. It conclusively demonstrates how the interaction between dietary components leads to metabolic outcomes that are more than the sum of their parts. Regarding the comments of reviewer 3:

1) All scientific experiments involve tradeoffs and limitations. An advantage of nutritional geometry experiments is exploring many diets; but in the absence of infinite funds, fewer genotypes and sexes of mice can be studied. While the authors could more clearly emphasize that this is a limitation of the current study - rather than just highlighting the need for further study, explicitly stating that the use of C57BL/6J males only is a limitation - but I do not view this study design choice as a problem.

2) Similarly, there are undoubtedly age-dependent effects of diet and studies in older animals might give somewhat different results - the focus on young animals can be acknowledged as a limitation - but 8 week old mice are fully developed young adults. If we want to compare to humans, well, human children are exposed to a similar range of diets as human adults, and obesity in many likely begins early in life. Many children and teenagers are exposed to HFCS and many are obese or overweight.

3) The authors made a bold and inspired choice to keep energy density constant; there would undoubtedly have been valid criticism had they allowed energy density to vary. Any diet study needs to make tradeoffs, this seems a reasonable choice to me given that carbohydrates and fats cannot be equally exchanged. I would likely use a similar approach in any future study of my own.

4) While I agree these types of measurements would be interesting, I agree with the authors that brain and physiological food intake should be going in the same direction, and I feel the proposed analysis is beyond the scope of the present study.

5) The update to the title is appropriate and should resolve most issues of concern in my opinion. Perhaps the abstract and introduction could also be slightly revised to focus on "THE METABOLIC EFFECTS OF DIETARY FAT, SUGARS AND FAT-SUGAR INTERACTION" rather than the "THE ENERGY BALANCE AND CARBOHYDRATE INSULIN MODELS OF OBESITY" - the opening of the abstract and introduction now start from a slightly different place than you would expect given the title. Only a suggestion.

In summary - mice are not humans, and that is a limitation of any mouse study. Thus this is NOT the very last word - much more remains to be done. However, the conclusions drawn by the authors from the results of the present study are appropriate, and while I feel they should do a slightly better job explicitly highlighting the limitations of the study in the "Discussion" section, and perhaps rephrasing abstract and info as mentioned, the overall message is extremely exciting. In conclusion, this is an innovative, technically demanding and well executed study, will be of wide interest, and I look forward to its acceptance and the discussion it will stimulate in the scientific and nutrition communities.

Responses to Nature Communications reviewers

REVIEWERS' COMMENTS:

Reviewer #1 (Remarks to the Author):

I appreciate the authors' thorough responses to my previous comments. However, it is still questionable whether their study has any meaningful implications on human diets and health consequences. The most concerning part is the fact that they used only solid fructose, which is known not as harmful as liquid fructose according to numerous human intervention studies. They raised technical issues of feeding mice with their special diet but they can at least try pair feeding or feeding restriction methods to overcome this issue. Ad libitum feeding is simple and easy for investigators but it limits the value of the study. This reviewer does not ask to repeat all their comprehensive experiments with liquid fructose, but the authors should perform a few critical experiments strategically at least to strengthen their previous conclusion or draw different conclusions. Also, I agree with the high cellulose content concern raised by another reviewer, which limits relevance to human diet. The authors mentioned they cannot measure postprandial metabolism without sacrificing mice, but measuring fed state metabolism is possible with simple blood tests at night times or early morning. Lastly, thermoneutral temperatures could affect many factors other than just brown fat effects such as eating behavior, obesogenic effect, insulin sensitivity, energy consumption, etc. How do authors know this without testing it on their dietary conditions? Overall, while this study is interesting and potentially important, this reviewer hopes to see data from experimental conditions that better mimic human diet and physiology to support the publication.

Response:

We thank the reviewer for providing feedback on our revised manuscript. We have provided our responses to these comments below.

Implications for human health:

In response to this reviewer's comments about the implications of our study to human diets and health, we agree with reviewer-4's comment that mice are not humans, and that is a limitation of any mouse study, but our conclusions are appropriate and the study itself is well executed.

Solid vs liquid fructose:

As we previously mentioned in our responses, we agree that fructose is more detrimental for metabolic health if consumed in liquid form. However, we decided to use solid experimental diets because our aim was to study the effects of dietary sugar-fat substitution on metabolic outcomes. It is very challenging to make mice consume large amounts of fats in liquid form (mice are unlikely to drink oils to the same extent as sugar solutions). Importantly, we were able to identify a fructose-glucose ratio (50:50) that was obesogenic even when these sugars were consumed in solid form. Nonetheless, we have added a sentence to the discussion highlighting the value of studying the metabolic effects of sugars in the beverage format.

Pair feeding:

Pair feeding is an experiment that could be useful, and we have added this suggestion in the limitations of the revised manuscript. Here, we would like to add that while pair feeding has its benefits, it also has its own limitations. In such experiments, the pair-fed group could become a case of time-restricted feeding (as the mice eat the limited amounts of food provided quickly and go without food for extended periods) which will have its own confounding effects on data interpretation.

High cellulose content:

Here would again quote the comment by reviewer-4: "The authors made a bold and inspired choice to keep energy density constant; there would undoubtedly have been valid criticism had they allowed energy density to vary. Any diet study needs to make tradeoffs, this seems a reasonable choice to me given that carbohydrates and fats cannot be equally exchanged. I would likely use a similar approach in any future study of my own."

Fed state metabolism:

We had already acknowledged this suggestion in the limitations of this study. However, repeating the whole in vivo experimental study again for the markers of fed state metabolism is not feasible at this stage.

Thermoneutrality:

We acknowledge this comment and we have added a sentence in the limitations of the study about the potential benefits of repeating our mouse experiments under thermoneutral conditions. We designed study to investigate the metabolic effects of dietary fat vs sugars in the context of the carb-insulin and energy balance models of obesity. We think that it is unlikely that the differences in the metabolic effects of fat vs sugar consumption would be drastically affected by housing mouse in thermoneutral environment. It may affect the magnitude of the observed differences, but it is not likely that the differences between metabolic effects of fat and sugar intake will change altogether by increasing the housing temperature.

Reviewer #2 (Remarks to the Author):

This is an interesting paper that investigates the role of monosaccharides versus fat with 18 different diets that are isocaloric in calories/g of food. All of the diets varied from 50 to 70 percent carbohydrate and from 10 to 30 percent fat, with 20 percent protein in all diets. The main finding was that energy intake was greatest with a 50/50 mixture of glucose and fructose, but this was overcome to some extent by increasing the fat content. The authors argue that this supports both the CHO insulin theory and the energy balance theory.

Comments

While I agree that the data does support both models, I do think there are shortcomings. The two major ones are that the studies involve only males and one strain of mouse, and that there are very limited studies investigating mechanisms. The increased energy intake is likely from leptin resistance but could involve other mechanisms and it would be important to understand this better. A third weakness is that all of the models involved high carb diet (50-70%). There is some evidence that high CHO diets may be able to stimulate energy intake by inducing leptin resistance which is not seen

with high fat diets, but in this case there were no controls with low (30% CHO) diets to really tease this out. I also agree with Reviewer 1 who argued that some postprandial studies would have been helpful, as well as my prior request to look at endogenous fructose pathways. Finally, it would have also been optimal to provide some mice with fructose/glucose in the drink, as that is much more obesogenic than putting it into the water.

Response:

In the previous round of revisions, we had already acknowledged that future research should focus on extending this study to females and other strains of mice. Furthermore, we also addressed the comments about leptin resistance, endogenous fructose pathways and administering sugars as drinks.

We did not use very low carbohydrate diets in this study because that did not give us enough room to study the impact of carbohydrate type and fructose-glucose ratios. Further, had we replaced more carbohydrates in the diets with fat, it would have required using very high amounts of cellulose to keep the diets isocaloric. This would have then attracted the criticism that the observed metabolic effects were due to high amounts of cellulose in the diet. Therefore, after evaluating the pros and cons of these options, we came up with the dietary compositions that allowed us to achieve the study objectives while also minimising the impact of confounding factors.

Reviewer #4 (Remarks to the Author):

The editor has asked me to comment on the author's response to Reviewer #3 due to my expertise in diet design and mouse studies. I have read the paper and the reviews, and I think that this manuscript by Simpson and colleagues is interesting, reports novel results that will be of interest to a broad audience of scientists, as well as to the public. It conclusively demonstrates how the interaction between dietary components leads to metabolic outcomes that are more than the sum of their parts. Regarding the comments of reviewer 3:

1) All scientific experiments involve tradeoffs and limitations. An advantage of nutritional geometry experiments is exploring many diets; but in the absence of infinite funds, fewer genotypes and sexes of mice can be studied. While the authors could more clearly emphasize that this is a limitation of the current study - rather than just highlighting the need for further study, explicitly stating that the use of C57BL/6J males only is a limitation - but I do not view this study design choice as a problem.

2) Similarly, there are undoubtedly age-dependent effects of diet and studies in older animals might give somewhat different results - the focus on young animals can be acknowledged as a limitation - but 8 week old mice are fully developed young adults. If we want to compare to humans, well, human children are exposed to a similar range of diets as human adults, and obesity in many likely begins early in life. Many children and teenagers are exposed to HFCS and many are obese or overweight.

3) The authors made a bold and inspired choice to keep energy density constant; there would undoubtedly have been valid criticism had they allowed energy density to vary. Any diet study needs to make tradeoffs, this seems a reasonable choice to me given that carbohydrates and fats cannot be equally exchanged. I would likely use a similar approach in any future study of my own.

4) While I agree these types of measurements would be interesting, I agree with the authors that brain and physiological food intake should be going in the same direction, and I feel the proposed analysis is beyond the scope of the present study.

5) The update to the title is appropriate and should resolve most issues of concern in my opinion. Perhaps the abstract and introduction could also be slightly revised to focus on "THE METABOLIC EFFECTS OF DIETARY FAT, SUGARS AND FAT-SUGAR INTERACTION" rather than the "THE ENERGY BALANCE AND CARBOHYDRATE INSULIN MODELS OF OBESITY" - the opening of the abstract and introduction now start from a slightly different place than you would expect given the title. Only a suggestion.

In summary - mice are not humans, and that is a limitation of any mouse study. Thus this is NOT the very last word - much more remains to be done. However, the conclusions drawn by the authors from the results of the present study are appropriate, and while I feel they should do a slightly better job explicitly highlighting the limitations of the study in the "Discussion" section, and perhaps rephrasing abstract and info as mentioned, the overall message is extremely exciting. In conclusion, this is an innovative, technically demanding and well executed study, will be of wide interest, and I look forward to its acceptance and the discussion it will stimulate in the scientific and nutrition communities.

Response:

We are pleased to know that this reviewer found our manuscript of great interest.

1. We have updated the manuscript to clearly state that using only C57BL/6J mice of male sex is a limitation of this study.
2. We agree that 8-week-old mice are sexually mature, and our study design helps in mirroring the exposure to HFCS in teenagers and adolescent humans.
3. One of the main aims of this study was to compare the metabolic effects of fat and sugars per se, so it was crucial to disentangle their metabolic effects from their calorie density. Therefore, it was important to use isocaloric diets and we are glad that this reviewer appreciates this strength of our study design.
4. We thank the reviewer for agreeing that studies on brain tissues are beyond the scope of this present manuscript.
5. We have modified the abstract and introduction as suggested by the reviewer.